# Energy Balancing and Lifetime Extension: A Random Quorum-Based Sink Location Service Scheme for Wireless Sensor Networks

**DOI:** 10.3390/s25134078

**Published:** 2025-06-30

**Authors:** Yongje Shin, Jeongcheol Lee, Euisin Lee

**Affiliations:** 1School of Information and Communication Engineering, Chungbuk National University, Cheongju 28644, Republic of Korea; yjshin@cbnu.ac.kr; 2Department of Supercomputing Acceleration Research, Korea Institute of Science and Technology Information (KISTI), Daejeon 34141, Republic of Korea; jclee@kisti.re.kr

**Keywords:** wireless sensor networks, sink location service, quorum, crossing point, energy balancing

## Abstract

Geographic routing is an appealing method for wireless sensor networks, as it routes data packets solely based on nodes’ location information rather than global network topology. A fundamental requirement for geographic routing is that source nodes must know the locations of sink nodes to deliver their data. To efficiently provide sink location information, quorum-based sink location service schemes have been introduced, using crossing points between sink location announcement (SLA) and sink location query (SLQ) quorums. However, existing quorum-based schemes typically construct quorums along fixed paths, causing rapid energy depletion of particular sensor nodes and resulting in shorter network lifetimes, especially in irregular sensor fields. To overcome this limitation, we propose an energy-efficient quorum-based sink location service scheme that extends network lifetime by reducing and balancing sensor nodes’ energy consumption. Specifically, our scheme constructs a quadrilateral-shaped SLA quorum using four randomly selected points, and a line-shaped SLQ quorum defined by two randomly chosen points located inside and outside the SLA quorum, respectively. We also address key issues of the proposed scheme, including network holes, irregular boundaries, multiple sources and sinks, and Base Zone sizing, and present methods to handle them. Simulation results demonstrate that the proposed scheme outperforms existing approaches, achieving approximately 29% lower total energy consumption and 27% higher energy balancing across sensor nodes on average.

## 1. Introduction

Wireless sensor networks (WSNs) have drawn significant attention due to their wide range of applications, including environmental monitoring, military surveillance, disaster detection, and smart agriculture [1]. These networks typically consist of a large number of low-power, battery-operated sensor nodes that cooperatively sense, process, and transmit data to designated sink nodes [2,3]. Since these sensor nodes are often deployed in remote or inaccessible areas, replacing or recharging their batteries is generally infeasible. As a result, energy efficiency has become one of the most critical design objectives in WSN protocol development [4]. In particular, the communication subsystem—responsible for data routing and dissemination—accounts for a major portion of energy consumption, making the design of energy-efficient routing protocols essential for ensuring long-term network operation and data reliability [5,6].

Among various routing strategies, geographic routing has emerged as an appealing approach for WSNs [7,8,9]. This is primarily because it forwards data packets based solely on local position information of neighboring and destination nodes, without requiring global topology information or complex routing tables. This localized decision making reduces control overhead, enhances scalability, and makes geographic routing highly suitable for large-scale and dynamic sensor networks. Moreover, the stateless nature of geographic routing eliminates the need for route maintenance, further conserving energy. However, a fundamental requirement of this approach is that source nodes must know the location of sink nodes before data transmission [10,11,12]. Without sink location information, data delivery cannot be properly initiated, leading to routing failures, increased latency, and unnecessary energy consumption. Therefore, an efficient sink location service (SLS) is essential to enable practical and reliable geographic routing in WSNs.

To provide sink location information efficiently, a variety of SLS schemes have been proposed, which aim to provide the location information of sinks to source nodes in a timely and energy-efficient manner. A straightforward method is network-wide flooding, in which a sink periodically broadcasts its location to all nodes in the network [13]. Although this guarantees full dissemination of sink information, it incurs significant energy consumption, especially in large-scale networks or in scenarios with multiple sinks. To reduce the overhead, localized flooding schemes such as TTDD limit the broadcast region by organizing the network into grid structures [14]. While such schemes reduce the number of transmissions, they still suffer from high setup and maintenance costs and are less adaptable to irregular or dynamic network topologies.

To avoid flooding for SLS, various quorum-based SLS schemes have been introduced [15]. These schemes leverage crossing points between SLA and SLQ quorums. In these schemes, an SLA packet from a sink and an SLQ packet from a source are forwarded along predefined quorum paths designed to guarantee at least one crossing point between them. The crossing point serves as the location where the source retrieves the sink’s information. One representative example is the column–row method, where SLA and SLQ messages travel in perpendicular directions (e.g., north–south and east–west). While simple, this method assumes a regular rectangular network and fails to guarantee a crossing point in real-world deployments with voids, concave areas, or irregular boundaries.

To support such irregular topologies, more advanced quorum-based schemes have been developed [10,11]. These schemes employ predefined geometric paths—such as circular or perimeter-based routes—for constructing quorums. For example, a perimeter-based scheme forwards messages toward and along the network boundary [10], while a circular quorum scheme centers the sink’s announcement along a fixed circular path [11]. These designs improve the probability of intersection between SLAs and SLQs in non-rectangular or void-rich sensor fields where traditional column–row methods often fail. However, they introduce challenges in energy efficiency and load balancing. Since SLA and SLQ messages are routed along deterministic, fixed paths, certain nodes are repeatedly burdened with message forwarding, leading to energy hotspots and early depletion. These fixed paths also limit scalability, as increasing numbers of sources and sinks can cause congestion and accelerate energy exhaustion. Thus, despite their geometric robustness, these methods fall short in achieving balanced energy usage and long-term network sustainability.

To address these limitations, we propose an energy-efficient and balanced quorum-based sink location service for geographic routing in irregular WSNs. The proposed scheme avoids fixed-path structures and instead constructs randomized quorums to evenly distribute traffic load across the network. Specifically, a quadrilateral-shaped SLA quorum is constructed by connecting four randomly selected nodes, ensuring spatial diversity and mitigating energy concentration. The corresponding SLQ quorum is formed as a straight line between two randomly selected nodes located inside and outside the SLA region, respectively, guaranteeing at least one crossing point without relying on a fixed route. To enhance the practicality of the proposed scheme, we also address key challenges in real-world deployments: (1) handling network holes and irregular boundaries that may disrupt quorum paths, (2) supporting multiple sources and sinks, and (3) determining an appropriate Base Zone size to avoid central congestion. For each of these, we present effective techniques to improve the reliability and scalability of the scheme. Finally, the proposed scheme is evaluated through extensive simulations under various network conditions. The results demonstrate that our scheme significantly outperforms existing quorum-based approaches, such as XYLS [10] and CLPS [11], in terms of both total energy consumption and energy balancing, ultimately extending the network lifetime and improving overall performance.

The rest of this paper is organized as follows. We examine the related works about sink location service in the literature in Section 2. Our energy-efficient quorum-based sink location service scheme is described in detail in Section 3. The considerations of the proposed scheme are provided to address its key challenges in Section 4. Simulation results are given for evaluating the performance of the proposed scheme in Section 6. Section 7 concludes this paper.

## 2. Related Works

Wireless sensor networks (WSNs) rely significantly on efficient routing and data dissemination techniques to ensure energy-efficient communication. A key challenge is minimizing energy consumption while maintaining energy balancing to prolong network lifetime. Geographic routing extensively uses the SLS, which provides sink location information to source nodes. An effective SLS must minimize energy usage and avoid localized depletion. Existing schemes fall into two main categories: flooding-based [13,14] and rendezvous-based approaches [10,11,16,17,18,19,20,21,22,23].

The flooding-based approach [13,14] disseminates sink location information through broadcasting and is further divided into network-wide flooding and localized flooding. In the network-wide flooding scheme [13], a sink broadcasts its updated location to all nodes in the network. This guarantees that every sensor node receives the location update but leads to significant energy inefficiency, as all nodes must forward messages. Nodes closer to the sink suffer from faster energy depletion, and the energy cost increases dramatically with the number of sinks, making this approach unsuitable for large-scale WSNs. To address these issues, Two-Tier Data Dissemination (TTDD) [14] was proposed as a localized flooding method. TTDD constructs a virtual grid from the source’s location and restricts flooding within specific grid cells. This reduces redundant transmissions and saves energy. However, the grid structure must be frequently updated as nodes deplete or change, and grid-line nodes face higher communication burdens, resulting in localized energy depletion.

To overcome the energy inefficiencies of flooding-based methods, rendezvous-based approaches have been introduced. These methods rely on predefined meeting points between SLA and SLQ. Rendezvous-based approaches are typically classified into hash-based and quorum-based schemes.

The hash-based approach uses geographic hash functions to efficiently store and retrieve sink location information [21,22,23]. Geographic Location Service (GLS) [21] employs a hierarchical structure where each node registers its location at multiple levels determined by hashing geographic regions. This reduces communication overhead and balances load, offering scalability for large and dynamic networks. Distributed Location Management (DLM) [22] extends GLS with improved scalability and robustness by optimizing registration and lookup under frequent network changes. Hierarchical Location Service (HLS) [23] divides the network into hierarchical regions and sub-cells, where nodes register and query locations using hash-based mapping. This design supports efficient location discovery, particularly in localized communication scenarios. Although hash-based schemes reduce flooding, uneven hash distribution can lead to query concentration and energy imbalance. Additionally, suboptimal mappings may cause redundant transmissions, increasing energy consumption.

To address these limitations, quorum-based approaches have been proposed [10,11,16,17]. In this method, SLA and SLQ messages are sent to predefined subsets of nodes, called SLA and SLQ quorums, respectively, which are designed to intersect at least once. A basic example is the column–row method, where a sink sends an SLA message in the north–south direction, while a source sends an SLQ message in the east–west direction. Though simple, this method guarantees intersection only in rectangular networks and fails in irregularly shaped or void-containing networks.

To support irregular network topologies, two types of quorum-based approaches have been explored: network perimeter knowledge-based and particular path-guided forwarding approaches. In the network perimeter knowledge-based approach, schemes such as NELS [16] and SLS-IR [17] use the location information of perimeter nodes. NELS selects anchor nodes on the network boundary, divides them into four quadrants, and sends SLAs and SLQs to opposite pairs to ensure intersection. While this reduces unnecessary querying, it introduces high control overhead due to flooding boundary information and causes anchor node energy imbalance. SLS-IR defines an inner rectangle within the network and applies the column–row quorum method inside it, reducing query space and enabling more predictable message paths. However, it also incurs high overhead from flooding the inner rectangle information and leads to energy burden on perimeter nodes of the inner region.

In contrast, the particular path-guided forwarding approach avoids the usage of flooding but constructs quorums only on sensor nodes of particular paths [10,11]. XYLS [10] forwards SLA and SLQ messages toward the perimeter and then along it in a clockwise direction, ensuring intersection. Although this method simplifies routing and reduces message overhead, it leads to energy imbalance and potential bottlenecks at perimeter nodes. An enhanced variant, CR+CR, constructs vertical and horizontal quorums for each SLA and SLQ, creating multiple intersections and improving robustness. However, this variant increases energy consumption and routing complexity, becoming less scalable as the number of sources and sinks grows. CLPS [11] constructs a circular SLA quorum centered at a predefined Base Node and a linear SLQ quorum from the source through the Base Node to the network perimeter. By ensuring an intersection between SLQ and SLA, CLPS significantly reduces control overhead and supports irregular network topologies by dynamically adapting paths around network holes. However, since SLA messages repeatedly traverse the same circular path, nodes along this path experience disproportionate energy consumption, causing energy imbalance and potential early node depletion.

Despite these various approaches, existing SLS schemes still struggle to simultaneously achieve energy efficiency and balanced energy consumption. Therefore, we propose a novel quorum-based SLS that avoids the use of perimeter information and forwarding. Our scheme constructs a lightweight structure composed of a quadrilateral SLA quorum and a line-based SLQ quorum, ensuring at least one crossing point between them. By randomly selecting four points on orthogonal lines for the SLA quorum and two endpoints for the SLQ quorum, the proposed scheme achieves both reduced energy consumption and balanced load distribution, offering an energy-efficient and scalable solution for geographic routing in WSNs.

## 3. Network Model and Scheme Overview

In this section, we present the network model and overview of the proposed scheme.

### 3.1. Network Model

As a network model of the proposed scheme, we consider a wireless sensor network in which a great number of sensor nodes are randomly deployed as the most common way of numerous applications. By this deployment way, irregular network shapes (i.e., not circles or rectangles) and void areas such as holes might be generated in the network as shown in Figure 1. Since wireless sensor networks can be used for various applications such as environmental surveillance, military operations, and object tracking, sensor nodes detect events related with used applications, makes a report data about the events, and sends the report data to sinks. In the proposed scheme, sensor nodes use a well-known geographic routing protocol such as GPSR and GDSTR to forward data packets between them in the network [7]. Basically, geographic routing needs the location information of three types to send packets. The first one is the location information of individual sensor nodes. We assume that every sensor node can obtain its own location information by using GPS [24] or localization schemes [25,26]. The location information of neighbor sensor nodes is the second one. Usually, every sensor node can obtain its neighbor nodes and their locations by periodically exchanging beacon messages with its location [27]. The last type of location information is the location information of destinations that are generally sinks in wireless sensor networks. In this paper, the proposed scheme is exploited to provide the location information of sinks to sensor nodes as sources. To implement our scheme, we also assume that every sensor node knows the center location of the network according to pre-installed information from a network administrator. A sensor node can know whether it is an edge node on the perimeter of the network by using a manual identification during the network deployment process or an automatic detection method [28,29] after finishing the network deployment.

### 3.2. Scheme Overview

The proposed scheme provides a sink location service for geographic routing in WSNs through a structured multi-phase process. Once sensor nodes are deployed in the field to monitor events of interest, the scheme begins its operation. Following deployment, the scheme initiates a *network initialization* phase to support sink location discovery in irregular sensor fields, including those with holes. During this phase, a predetermined initialization node selects a *Base Node*, as illustrated in Figure 1, which then constructs a *Base Zone* at the center of the network, as shown in Figure 2. This zone serves as a central coordination region for quorum construction.

After initialization, sink and source nodes proceed with their respective operations to enable an energy-balanced, quorum-based sink location service, as depicted in Figure 3. The primary goal of the scheme is to ensure at least one crossing point between the *sink location announcement (SLA)* quorum of a sink and the *sink location query (SLQ)* quorum of a source, while also distributing energy usage evenly among sensor nodes.

To guarantee this crossing point, the proposed scheme leverages the following geometric principle:**Crossing Lemma:** Given a quadrilateral and a line segment, if one endpoint of the line lies inside the quadrilateral and the other lies outside, then the line must intersect the boundary of the quadrilateral at least once. Thus, the quadrilateral and the line have at least one crossing point between them.This principle is supported by the following mathematical model:***Mathematical Model for Crossing Lemma***: Let the *SLA quorum* be represented as a simple, closed quadrilateral $Q\subset R^{2}$ formed by connecting four randomly selected points $\left\{q_{1},q_{2},q_{3},q_{4}\right\}$. The quadrilateral region *Q* induces a *Jordan domain*, meaning it is a bounded open subset of $R^{2}$ whose boundary is a simple closed curve. Let ∂Q denote the *boundary* of *Q*, formally defined as:(1)$$\partial Q=\overset{4}{\underset{i=1}{⋃}}\bar{q_{i}q_{i+1}},\;\mathrm{with}\;q_{5}=q_{1}.$$This represents the union of the four line segments forming the closed polygon. Let int(Q) denote the *interior* of the quadrilateral, i.e., the set of points strictly inside the polygon. And let the *SLQ quorum* be a line segment $L=\bar{s_{i}s_{o}}$, where: $s_{i}\in \mathrm{int}\left(Q\right)$ is a point randomly chosen *inside* the quadrilateral, $s_{o}\in R^{2}\setminus \bar{Q}$ is a point randomly chosen *outside* the closed region. Then, by the *Jordan Curve Theorem*, since the endpoints of *L* lie on opposite sides of the boundary ∂Q, it follows that there exists at least one intersection point:(2)L∩∂Q≠∅.This guarantees that the SLQ quorum intersects the SLA quorum at least once.

In the proposed scheme, each sink transmits its location information to the Base Zone. Using this information, the Base Zone constructs a quadrilateral SLA quorum by connecting four randomly selected nodes located outside the Base Zone. On the other hand, each source constructs a line-shaped SLQ quorum that connects a randomly selected node inside the Base Zone to another node near the network boundary.

Due to the spatial relationship between the two quorums, the crossing lemma ensures that they intersect at least once. At the intersection node, which receives both SLA and SLQ messages, the sink’s location is forwarded to the source as a sink location reply. With this information, the source can initiate geographic routing to deliver its data packet to the sink.

## 4. The Proposed Scheme

In this section, we explain the proposed scheme in detail. As examined in the overview of our scheme, it consists of four phases. Thus, we first present the phase of the network initialization fundamentally configured to achieve our energy-balanced quorum-based sink location service in Section 4.1. Next, two phases for the sink location announcement and sink location query are provided to construct a quadrangle quorum and a line quorum in Section 4.2 and Section 4.3, respectively. Last, the phase for sink location service is described to guarantee the crossing point between the quadrangle quorum and the line quorum, and to send sink location reply to sources in Section 4.4.

### 4.1. Network Initialization

After the network deployment of sensor nodes in a wireless sensor network, a random sensor node called a predetermined initialization node *IN* selects a sensor node called *Base Node* which is closest to the center location of the network. Generally, the predetermined initialization node is chosen and programmed by the operator of the network before the deployment. Since the network operator also knows the location and the coverage of the sensor network, it can know the center location of the network and send the center location to all sensor nodes before their deployment. Figure 1 shows the network deployment of sensor nodes and the *Base Node* selection process in a irregular wireless sensor network with a hole. For robust *Base Node* allocation, the proposed scheme exploits the modified Perimeter Refresh Protocol (PRP). The initialization node generates a *Base Node Selection* message with the location information of the network center and sends it toward the network center by geographic routing. The message arrives at the sensor node that includes the network center within its one hop coverage and does not have any neighbor sensor node closer to the network center than it. Then, by using perimeter mode routing, the sensor node sends a *Replica Selection* message toward edge sensor nodes on a planar graph that contains the network center inside. After receiving the *Replica Selection* message, the edge nodes become replica nodes, and the closest one to the network center among the replica nodes is selected as a Base Node. Because the *Base Node* is also a sensor node, if it fails due to several reasons such as energy exhaustion and device breakdown, one of the replica nodes tries to find a new *Base Node* by using the same perimeter mode routing. In this way, the proposed scheme can robustly allocate a Base Node in the network center without any global flooding.

The proposed scheme needs to differently construct a quorum of sink location announcements by each sink to achieve the balanced energy consumption of sensor nodes. To do this, the Base Node needs to know the location information of the four edge sensor nodes on the perimeter of the sensor network from its own location. Thus, after the process of a *Base Node* selection, the Base Node initializes an *Edge Location Request (ELR)* packet with its location information and sends the *ELR* packet to the four edge sensor nodes on horizontal (i.e., east and west) and vertical (i.e., north and south) lines from itself by geographic routing in order to obtain their location information. For each of the four edge sensor nodes, the destination field of the ELR packet in geographic routing is set to (XB, +∞), (+∞, YB), (XB, −∞), and (−∞, YB), respectively, where (XB, YB) is the location of the Base Node. If the ELR packet is received by an edge sensor node on the network perimeter, it replies with its location information to the Base Node.

Figure 2 shows the process to obtain the four edge sensor nodes from a *Base Node* and to configure a Base Zone around the network center. In Figure 2, the thick dotted curved line indicates the perimeter of the irregular wireless sensor network. White nodes represent edge sensor nodes on the network perimeter. To facilitate discussion, all general sensor nodes in the network are not drawn out in this figure. In Figure 2, the four-way edge nodes (i.e., the north edge node E1, the east edge node E2, the south edge node E3, and the west edge node E4) send their own location information to the Base Node B if they receive an *ELR* packet from the *Base Node*. Since the Base Node should receive a number of control packets (for example, packets for replying to ELR packets by the four-way edge nodes, packets for sending the sink location announcement by sinks, and packets for sending the sink location query by sources), these control packets cause much congestion around the Base Node. To prevent this congestion problem, the proposed scheme configures a restricted region called *Base Zone* around the *Base Node* as shown in Figure 2. The sensor nodes in the *Base Zone* conduct the functions of the *Base Node* on behalf of it when they receive control packets toward the *Base Node*. To inform the Base Zone about the sensor nodes within it, the *Base Node* floods a *Base Zone Announcement* packet with its location information into the Base Zone. If the sensor nodes in the Base Zone receive the *Base Zone Announcement* packet, they can save the location of the *Base Node* and conduct the functions of the Base Node. The length of the Base Zone, α, is predefined by the network operator; however, it can be reduced according to the distance between the four-way edge nodes and the Base Node. When the network initialization is completed, our quorum-based sink location service scheme performs the sink location announcement, the sink location query, and the sink location service according to Algorithm 1. Table 1 shows the explanation of parameters used in Algorithm 1.
**Algorithm 1** Quorum-Based Sink Location Service  1:**procedure** SLA(*S*)              ▹ on requesting event data at sink *S*  2:    dests←ComputeQuadDestinations(BaseNode,α)  3:    rem←|dests|  4:    packet←{type=LSLA,S_loc=S.loc,int=S.int}  5:    **while** rem>0 **do**  6:        X←currentNode()  7:        X.LSLA_info←packet  8:        nextDest←dests[0]  9:        dests←rotateLeft(dests)10:       rem←rem−111:       geo_forward(packet,nextDest)12:    **end while**13:**end procedure**14:**procedure** SLQ(N,interest)              ▹ on detecting an event at node *N*15:    pkt←{type=SLQ,N_loc=N.loc,int=interest}16:    geo_forward(pkt,BaseZonePerimeter)17:    geo_forward(clone(pkt),NetworkEdge)18:**end procedure**19:**procedure** SLR(*X*)              ▹ on receiving LSLA or SLQ packets at node *X*20:    **if** packet.type=LSLA **then**21:        X.LSLA_info←packet22:    **else if** packet.type=SLQ **then**23:        X.SLQ_info←packet24:    **end if**25:    **if** X.LSLA_info.int=X.SLQ_info.int **then**26:        reply←{type=SLR,S_loc=X.LSLA_info.S_loc}27:        geo_forward(reply,X.SLQ_info.N_loc)28:    **end if**29:**end procedure**

### 4.2. Sink Location Announcement

As shown in Figure 3, when a sink S1 wants to receive data from sources in the proposed scheme, it initializes an SLA packet including its location, interest, and interest duration information, and sends the SLA packet to the center location of the network by geographic routing to construct a quorum of the SLA packet. The procedure SLA(S) of Algorithm 1 presents the process to construct an SLA quorum in the sink. As a result of geographic routing to the network center, a sensor node on the Base Zone consequently receives the SLA packet and then sends the SLA packet to the Base Node B because it knows the location information of the Base Node. Thus, the Base Node B receives the SLA packet. Then, the Base Node initializes a line sink location announcement (LSLA) packet including the following fields: Destination, Routes, Sink Location, Sink Interest, Duration, and Metric to construct a quadrilateral quorum for S1. The routes field represents a list of four intermediate destinations (IN Destinations) for a path for the LSLA packet to construct an LSLA quadrilateral quorum. To balance the energy consumption of sensor nodes, the proposed scheme constructs an LSLA quadrilateral quorum for every sink in different locations. Thus, to distribute paths of LSLA packets from sinks in the proposed scheme, four intermediate destinations (IN Destinations) are randomly selected on four lines (north, east, south, and west lines) from the Base Node, respectively, and are calculated as follows:(3)$$\begin{array}{c} \;NorthIN_{D}estination:\left(X_{B},R\left(Y_{B}+\frac{\alpha }{2},Y_{E1}\right)\right),\\ \;EastIN_{D}estination:\left(R\left(X_{B}+\frac{\alpha }{2},X_{E2}\right),Y_{B}\right),\\ \;SouthIN_{D}estination:\left(X_{B},R\left(Y_{E3},Y_{B}-\frac{\alpha }{2}\right)\right),\\ \;WestIN_{D}estination:\left(R\left(X_{E4},X_{B}-\frac{\alpha }{2}\right),Y_{B}\right), \end{array}$$
where the function *R(A, B)* represents a random value between *A* and *B*. The Sink Location, the Sink Interest, and the Interest Duration fields in the LSLA packet are filled with the information of sink location, interest, and interest duration in the received SLA packet, respectively. The Metric field is set to five because the LSLA quadrilateral quorum needs to finish its construction with only 360 degrees.

When four intermediate destinations are calculated for north, east, south, and west, respectively, the Base Node sets the Destination field of the LSLA packet to the closest one (i.e., north for the sink S1 in Figure 3) among the four IN Destinations to itself and the routes field to east, south, west, and north for the clockwise direction. Then, it sends the LSLA packet to the location of the closest IN Destination by geographic routing. In the sensor network, when a sensor node receives the LSLA packet, it saves the location and interest of the sink S1 in the LSLA packet to its sink information table during the Duration seconds, and then forwards the LSLA packet to a next-hop node to the closest IN Destination location according to the same rule. However, especially if the Metric is five, sensor nodes receiving the LSLA packet never save the information in the LSLA packet. Moreover, if the Metric is five, an LSLA packet can be forwarded in the Base Zone. If not, an LSLA packet would be detoured according to the perimeter of the Base Zone.

When the LSLA packet of S1 arrives at the sensor node R1 that is the closest node to the north intermediate destination, R1 changes the Destination field in the LSLA packet as the next IN Destination (for example, the east intermediate destination in Figure 3) and reduces the value of the Metric by one. Then, R1 sends the LSLA packet to the next IN Destination (i.e., east). This process is continued until the value of the Metric becomes zero. Eventually, the LSLA packet is returned to R1. In other words, if the Metric value is zero, it means that the LSLA packet has successfully traveled around the whole sensor network, which is a quadrilateral quorum R1R2R3R4 constructed by four random locations and includes the Base Zone inside itself. In this way, the proposed scheme can dispersively construct quadrilateral quorums of the SLA for sinks in the sensor network in order to achieve the balanced energy consumption of sensor nodes.

### 4.3. Sink Location Query

When a sensor node N1 detects an event and becomes a source node as shown in Figure 3, it initializes an SLQ packet including the information of the source location and the event type for sending it to sinks in a wireless sensor network. The procedure SLQ(N, interest) of Algorithm 1 presents the process to construct an SLQ quorum at the source node. To make a crossing point between an SLA quorum and an SLQ quorum for the sink location service, the proposed scheme allows the source node to construct a line quorum of its SLQ packet between the perimeter of the Base Zone and an edge of the sensor network perimeter in the sensor network. To do this, N1 first sends a copy of the SLQ packet toward the center location of the network by geographic routing on the path N1Z1 in Figure 3. However, if a sensor node that is in the Base Zone and is close to the perimeter of the Base Zone receives the SLQ packets, it then stops sending the SLQ packet. This policy prevents the congestion and collision problem due to SLQ packets in the Base Zone.

To accomplish the construction of the line quorum, N1 also sends another copy of the SLQ packet to the farthest neighbor node from the center location of the network other than itself in order to deliver the SLQ packet to an edge node on the network perimeter. During this process, if an intermediate node cannot find a farther neighbor node as the next-hop node toward the edge node, it forwards the SLQ packet to its previous intermediate node. The previous intermediate node marks the node and reforwards the SLQ packet to the second farthest neighbor node. This process continues until a network edge node (for example, the edge node E5 in Figure 3) receives the SLQ packet. Thus, the path N1E5 of the SLQ packet is constructed in the sensor network. When a sensor node in the network receives the SLQ packet, it saves the source location and the event type in the SLQ packet to its source information table. As a result, the line quorum of the SLQ packet for the source node N1 is constructed on the line Z1N1E5 to provide the sink location service of the proposed scheme as shown in Figure 3. Because the proposed scheme makes every source node construct a line quorum of its SLQ packet from its own location, line quorums of all source nodes are dispersively constructed to different locations, and thus, the energy consumption of sensor nodes is balanced.

### 4.4. Sink Location Service

Our sink location service scheme provides a quorum-based sink location service for geographic routing in irregular wireless sensor networks. Basically, a quorum-based sink location service relies on a crossing point between an SLA quorum of a sink and an SLQ quorum of a source node. To guarantee a crossing point between an SLA quorum and an SLQ quorum, the proposed scheme is supported by the following theorem and proof.

***Crossing Lemma for SLS***: Given a quadrilateral quorum and a line quorum in a two-dimensional irregular sensor field, if one end of the line quorum is inside the quadrilateral quorum and the other end of the line quorum is outside of the quadrilateral quorum, then the quadrilateral and the line quorums have at least one crossing point.

***Proof***: According to the above theorem, our scheme guarantees that a quadrilateral quorum of an LSLA packet for a sink and a line quorum of an SLQ packet for a source node have at least one crossing point. To do this in the proposed scheme, a quadrilateral LSLA quorum is constructed within the irregular sensor network and outside of the Base Zone, and has a closed form. On the other hand, a line SLQ quorum is constructed between the perimeter of the Base Zone and the edge of the network perimeter, and cuts between the Base Zone and the network perimeter. Then, one end of the line quorum is the perimeter of the Base Node within the quadrilateral quorum, and the other end of the line quorum is on the edge outside of the quadrilateral quorum.

Figure 3 shows an example in which the proposed scheme provides one crossing point between an LSLA quadrilateral quorum and an SLA line quorum in an irregular sensor field according to our theorem and proof. As shown in Figure 3, the LSLA packet delivery path R1R2R3R4 for the sink S1 is located between the Base Zone and the network perimeter. Then, one end of the SLQ packet delivery path Z1N1E5 for the source node N1 is the Base Zone that is inside the LSLA packet delivery path R1R2R3R4 for the sink S1, and the other end of the SLQ packet delivery path is the edge node E5 that is outside the LSLA packet delivery path. Thus, the proposed quorum-based sink location service scheme can guarantee that the LSLA path and the SLQ path have at least one crossing point, e.g., the black square node C1.

As a result of our SLS scheme, the sensor node that is located on the crossing point C1 receives both the LSLA packet of the sink S1 and the SLQ packet of the source node N1 as shown in Figure 3. From the LSLA and SLQ packets, the sensor node can know about the information of the source location, the event type, the sink location, and the sink interest. Then, if the event type matches the sink interest, the sensor node makes a sink location reply (SLR) packet with the sink location information to provide a sink location service to the source node. The procedure SLA(X) of Algorithm 1 presents the process to send an SLR packet to the sensor node on the crossing point. Next, the source node sends the SLR packet to the source node by geographic routing on the dotted curve line C1N1 as shown in Figure 3. Consequently, the source node receives the SLR packet and sends its data packets to the sink by geographic routing. This is the basic idea of the proposed sink location service scheme.

## 5. Considerations of the Proposed Scheme

In this section, we address considerations of the proposed scheme to make its implementation practical. We first explain how our scheme handles holes and irregular network boundaries. Next, we show how our scheme supports multiple sinks and sources. Additionally, we examine how the size of the Base Zone affects our scheme. Last, we present how our scheme can be extended to support the mobility of sinks and source nodes.

### 5.1. Handling Holes and Irregular Network Boundaries

As LSLA packets and SLQ packets in the proposed scheme might meet holes in the process of geographic routing, the proposed scheme also needs to conduct a recovery mode to make a detour around holes using a recovery scheme as shown in [30]. By using the recovery scheme, the proposed scheme can make a detour around the holes in order to complete the construction of quadrilateral quorums and line quorums as shown in Figure 4. Since the LSLA packet of a sink such as S1 meets a hole, it makes a detour around the hole. Even though the LSLA packet does not have a real quadrangle due to it making a detour around the hole, the quadrangle is not cut and opened. The SLQ of a source node such as N2 does not have a real straight line due to it bypassing a hole. However, the line is also not cut and opened.

When the proposed scheme selects four points to make a quadrangle quorum of an LSLA packet, any lines of the quadrangle quorum can be made along irregular network boundaries according to the positions of the four points. However, the quadrangle quorum is also not cut and opened. On the other hand, SLQ packets are not affected by irregular network boundaries. Since an SLQ packet arrives at a network perimeter, the SLQ is not forwarded further regardless of whether the network perimeter is regular or irregular. Thus, although LSLA and SLQ packets in the proposed scheme meet holes and an irregular network perimeter, the proposed scheme can make closed quadrangle quorums of LSLA packets and closed line quorums of SLQ packets. As a result, the proposed scheme can guarantee a crossing point C3 between the quadrangle quorum of the LSLA packet of a sink S1 and the line quorum of the SLQ packet of a source node N2. Thus, the sensor node, which is located on the crossing point C3, received both LSLA and SLQ packets, and can inform the source node N2 about the location information of the sink S1 by sending an SLR packet on the solid curve line C3N2.

### 5.2. Supporting Multiple Sources and Sinks

Multiple source nodes and sinks can be deployed in the sensor network simultaneously according to application scenarios of wireless sensor networks [11]. In such cases, the proposed scheme guarantees at least one crossing point between the quadrilateral SLA quorum of any sink and the line SLQ quorum of any source. As illustrated in Figure 4, once a sink S1 sends its SLA packet to the Base Node B, which then constructs a quadrilateral path R1R2R3R4, all source nodes (N1, N2) can retrieve the location of the sink from the nodes located on corresponding crossing points (C1, C3) on the LSLA path. Conversely, when a source N1 sends its SLQ packet, it can obtain the location information of all sinks (S1, S2) from the nodes located on the corresponding crossing points (C1, C2).

If a source node receives multiple pieces of sink location information from the corresponding crossing point nodes, it can send its data packets to each of the sinks by geographic routing. Then, the proposed scheme can exploit geographic multicast routing to efficiently send data packets from a source node to multiple sinks [31,32]. Specifically, geographic multicast routing enables energy-efficient data distribution from one source to multiple sinks based on their location information. Furthermore, our scheme supports scenarios where a source needs to deliver data to just a single sink among multiple available sinks. In this case, from an anycasting perspective, the source node can select one sink based on geographic distance (e.g., the closest sink) and subsequently send its data only to the selected sink using geographic unicast routing [33]. This selective transmission significantly reduces unnecessary energy consumption compared to multicasting when data from the source node is required by only one sink among multiple possible recipients.

Thus, our proposed quorum-based sink location service scheme robustly and efficiently accommodates multiple sources and sinks by supporting both location-based multicast and anycast routing according to application-specific data delivery requirements.

### 5.3. Size of Base Zone

The proposed scheme uses a Base Zone to prevent the Base Node from experiencing congestion from a lot of packet traffic such as SLA and SLQ packets. Then, the size (the length α) of the Base Zone influences to the lengths of SLAs and SLQs. If the length α is larger, the lengths of SLAs are shorter because SLAs are forwarded to only the perimeter of the Base Zone. On the other hand, if the length α is larger, the lengths of SLQs are longer because SLQs have more possibilities to detour the perimeter of the Base Zone. Thus, it is important to find an appropriate length α of the Base Zone to enhance the performance of the proposed scheme.

To determine an appropriate size (the length α) of the Base Zone and enhance the performance of the proposed scheme, we recommend a simulation-based analysis similar to that presented in Figure 10 of this paper. Specifically, various values of α should be tested under representative network scenarios, measuring performance metrics such as energy consumption, node lifetime, and routing overhead. As shown in Figure 10, an optimal α value can be identified by selecting the point at which the highest number of alive sensor nodes is achieved after repeated simulation rounds. Such empirical evaluation ensures a balanced trade-off between SLA and SLQ path lengths, ultimately reducing energy consumption and preventing rapid energy depletion of specific sensor nodes, thus achieving improved energy balancing.

## 6. Performance Evaluation

In this section, we compare our quorum-based location service scheme with XYLS [10] and CLPS [11] for performance evaluation using simulations. We first describe our simulation model and performance evaluation metrics. We next evaluate the performance of the proposed scheme, XYLS, and CLPS through simulation results.

### 6.1. Simulation Model and Performance Evaluation Metrics

We implemented XYLS, CLPS, and the proposed scheme in the QualNet network simulator 4.0 [34]. The model of sensor nodes follows the specification of MICA2 [35]. The transmission range of the sensor nodes is, omnidirectionally, 10 m, and their transmitting and receiving energy consumption rates are 49 mW and 29 mW, respectively. We use the IEEE 802.15.4 standard [36] as the MAC protocol for transmitting packets such as SLA, LSLA, SLQ, and data packets. As a default setting, we consider a sensor network of 1000 m × 1000 m where 400 sensor nodes are randomly deployed. In our simulation, one sink and one source are randomly located in the network every 100 s. In every scheme, the source and the sink send sink location query and announcement messages every 100 s, respectively. We define every 100 s as a sink location service round. As the default value for the length of the Base Zone in the proposed scheme, we use 100 m. The simulation time is 1000 s. To evaluate the performance of the proposed scheme, we use two performance evaluation metrics, energy consumption (i.e., the number of total transmission packets) and energy balancing (i.e., the number of alive sensor nodes) for the sink location service. In our simulation results, each data point is the average of 100 instances, and the confidence interval of 95 percent was applied accordingly.

### 6.2. Energy Consumption for the Number of Sinks and Sources

Figure 5 shows the energy consumption for the number of sinks. The x-axis represents the number of sink nodes (unit: nodes), and the y-axis indicates energy consumption, which is measured in terms of the total number of transmitted packets (unit: packets). As the number of sinks increase, XYLS consumes more energy than CLPS and the proposed scheme. In XYLS, every SLA quorum should be constructed along the network perimeter. However, CLPS and the proposed scheme do not require any perimeter forwarding. CLPS and the proposed scheme show similar graph slopes because SLA and SLQ distances are almost the same on average. However, the proposed scheme consumes less energy than CLPS due to the different cost of their network initialization phases. For network initialization, since XYLS does nothing, it has the minimum cost. In CLPS, the Base Node floods its location information to all sensor nodes in order to enable them to calculate their own circle height in the network. On the other hand, the Base Node in the proposed scheme just floods its location information into the small restricted area, the Base Zone. Although the proposed scheme additionally requests the location information of the four-way edge nodes, the flooding costs of CLPS are much bigger than those of the proposed scheme.

Figure 6 shows the energy consumption for the number of source nodes. The x-axis represents the number of source nodes (unit: nodes), and the y-axis indicates energy consumption, which is measured in terms of the total number of transmitted packets (unit: packets). XYLS consumes more energy than CLPS and the proposed scheme. The main reason is that a SLQ path of XYLS is twice as long as SLQ lines of CLPS and the proposed scheme in a worst case. However, due to high cost of the network initialization phase in CLPS, XYLS shows better performance than CLPS until the number of source nodes is 10. The proposed scheme consumes less energy than XYLS and CLPS because it has short SLQ paths (from a network edge to a perimeter node of the Base Zone) and low cost of the network initialization phase (restricted flooding within the Base Zone).

### 6.3. Energy Balancing for the Number of Sinks and Sources

Figure 7 shows the energy balancing for the number of sinks. The x-axis represents the number of sink nodes (unit: nodes), and the y-axis indicates energy balancing, which is measured in terms of the total number of alive sensor nodes (unit: nodes). As a performance metric for the energy balancing, we use the number of live sensor nodes after repeating simulation rounds. In this simulation scenario, the simulation round number is set to 500. As the number of sink nodes increases, XYLS greatly reduces the number of alive sensor nodes because all SLA packets are sent along all of the network perimeter nodes whose energy is rapidly exhausted due to excess energy consumption. Since CLPS allows SLA packets to be sent along their own designated circle paths, it has more live sensor nodes than XYLS. However, since sensor nodes on the designated circle paths suffer from excessive energy consumption, CLPS also quickly reduces the number of live sensor nodes. On the other hand, the proposed scheme allows each SLA packet to be sent along a quadrilateral path consisting of four random points. As a result, it balances the energy consumption of sensor nodes and thus has more live sensor nodes than XYLS and CLPS.

Figure 8 shows the energy balancing for the number of source nodes. The x-axis represents the number of source nodes (unit: nodes), and the y-axis indicates energy balancing, which is measured in terms of the total number of alive sensor nodes (unit: nodes). XYLS has fewer alive sensor nodes than CLPS and the proposed scheme because it makes all SLQ packets be sent along the network perimeter nodes. CLPS and the proposed scheme show a similar performance. However, the performance gap between CLPS and the proposed scheme becomes wider as the number of source nodes increases. The reason is related to the Base Node. In CLPS, all SLQ packets are sent to the Base Node, and thus, sensor nodes around the Base Node rapidly exhaust their energy. However, since the proposed scheme exploits the Base Zone, it distributes SLQ packets around the Base Zone.

### 6.4. Energy Balancing for the Number of Sources and the Size of Base Zone

Figure 9 shows the energy balancing for the number of sensor nodes. The x-axis represents the number of sensor nodes (unit: nodes), and the y-axis indicates energy balancing, which is measured in terms of the total number of alive sensor nodes (unit: nodes). Each scheme has 10 sources and 10 sinks, and the simulation round is set to 500. As the number of sensor nodes increases, each scheme has higher energy balancing. However, the slopes of these schemes are quite different. In XYLS, since all network perimeter nodes should cost much energy due to the SLA traveling independently of the density of the network, the increase is smaller than the other schemes. The graphs of CLPS and the proposed scheme seem to have a similar increasing rate; however, sensor nodes in the proposed scheme have increased their survival rate from 65 to 92 percent, while sensor nodes in CLPS have increased their survival from 43 to 76 percent.

Figure 10 shows the energy balancing for the length of the Base Zone in the proposed scheme. The x-axis represents the length of the base zone (unit: m), and the y-axis indicates energy balancing, which is measured in terms of the total number of alive sensor nodes (unit: nodes). In this scenario, the numbers of sources and sinks are set to 10, respectively. The number of sensor nodes is 400. After 500 rounds of simulation, we found that it shows the best performance at 100 m. As the length of the Base Zone becomes longer, more nodes can survive. The reason is that the length of the SLQ line is in inverse proportion to the size of the Base Zone. On the other hand, too large a Base Zone might lead to a frequent detour problem of LSLA packets, i.e., R5R6 in Figure 1. In that case, the energy of sensor nodes on the perimeter of the Base Zone is rapidly drained. If the nodes die, the detour route becomes longer as well as SLQ packets also possibly being detoured.

### 6.5. Energy Balancing for Network Irregularity

In Figure 11 and Figure 12, we compare the performance of these schemes for the network irregularity. The network irregularity is considered in terms of two cases, the irregularity degree of the network perimeter and the number of void areas. Figure 11 shows the energy balancing for the irregularity degree of the network perimeter. The irregularity degree of the network perimeter is defined as the ratio of the total perimeter length of the real network to the total perimeter length of the network of the circle. In the simulation, we consider the network of the circle with a radius 500 m. In Figure 11, as the irregularity degree of the network perimeter increases, the total length of the network perimeter also increases. Since XYLS is hardly affected by the distance of the network perimeter, it shows poor energy balancing. In CLPS, some SLA packets may travel along the partial network perimeter, especially in a large height circle. If there is another SLA packet that has a larger height than the circle, the energy of the sensor nodes of the partial network perimeter would be drained fast. However, since SLA packets of the proposed scheme stochastically travel along the network perimeter nodes, the scheme shows better energy balancing in terms of the irregularity degree of the network perimeter than other schemes.

Figure 12 shows the energy balancing for the number of void areas in the sensor networks. We consider circle regions of radius 10 m as the size of void areas. As the number of void areas increases, all schemes increase the energy consumption for the sink location service because SLA and SLQ packets might meet more void areas and thus must detour around the void areas. XYLS and the proposed scheme have a shorter delivery distance of SLA packets than CLPS because in CLPS, since an SLA packet travels around its own circle, detoured packets should turn back to its circle even if they have passed void areas. Therefore, the void detouring distance of CLPS is longer than other schemes. XYLS and the proposed scheme show similar performance. However, due to the high cost of the network perimeter traveling of XYLS, the proposed scheme shows better energy balancing than XYLS.

To further validate the effectiveness of the proposed scheme, we present a comparative performance analysis against two existing approaches XYLS and CLPS as summarized in Table 2. The results, averaged over 100 simulation runs, show that the proposed scheme significantly reduces energy consumption, with only 9000 ± 350 transmitted packets compared to 17,500 ± 520 and 13,000 ± 430 in XYLS and CLPS, respectively. In terms of energy balancing, the proposed method also achieves the highest number of alive sensor nodes (230 ± 10), indicating a more uniform distribution of energy usage and improved network longevity.

## 7. Conclusions

In this paper, we proposed an energy-efficient quorum-based sink location service scheme to extend the operational lifetime of WSNs in irregular deployment environments. Different from existing schemes, such as XYLS and CLPS, our scheme eliminates the need to flood the network perimeter and instead ensures a guaranteed intersection between a quadrilateral SLA quorum and a line SLQ quorum based on geometric principles. Then, our scheme selected four random points to construct the quadrilateral SLA quorum and two random points to construct the line SLQ quorum to reduce and balance the energy consumption of sensor nodes. Simulation results confirmed that our scheme significantly lowers total energy usage and enhances energy balance, demonstrating its practical value in energy-constrained environments such as remote monitoring or military surveillance. Specifically, the proposed scheme achieved approximately 29% lower energy consumption and 27% better energy balancing compared to XYLS and CLPS. By distributing the communication load more evenly, the scheme supports sustainable and fair operation across diverse network conditions. Moreover, the integration of quorum theory with geographic routing introduces a lightweight yet robust framework adaptable to irregular topologies. This theoretical and practical synergy underscores the applicability of our method for real-world WSN deployments and highlights its contribution to the design of scalable, energy-aware sensor network protocols. 

## Figures and Tables

**Figure 1 sensors-25-04078-f001:**
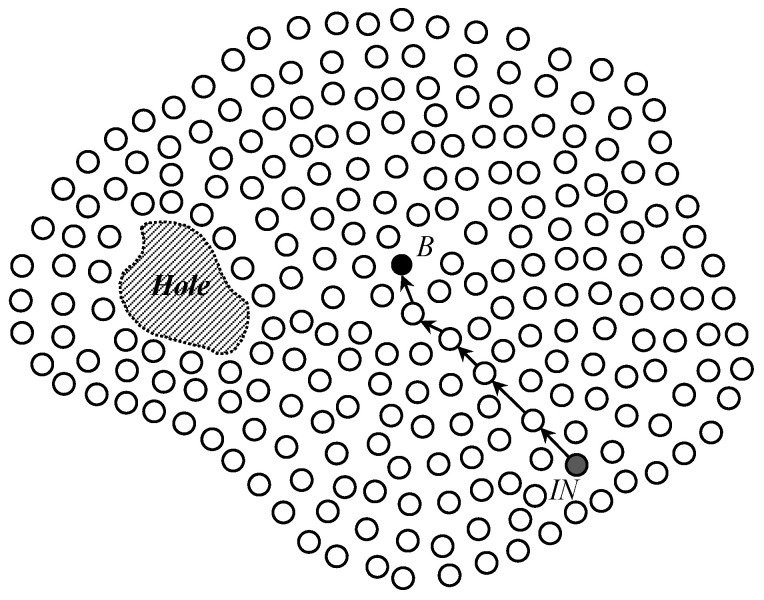
The construction of an irregular wireless sensor network with a hole by the network deployment of sensor nodes and the selection of the *Base Node B* by a predetermined initialization sensor node *IN*.

**Figure 2 sensors-25-04078-f002:**
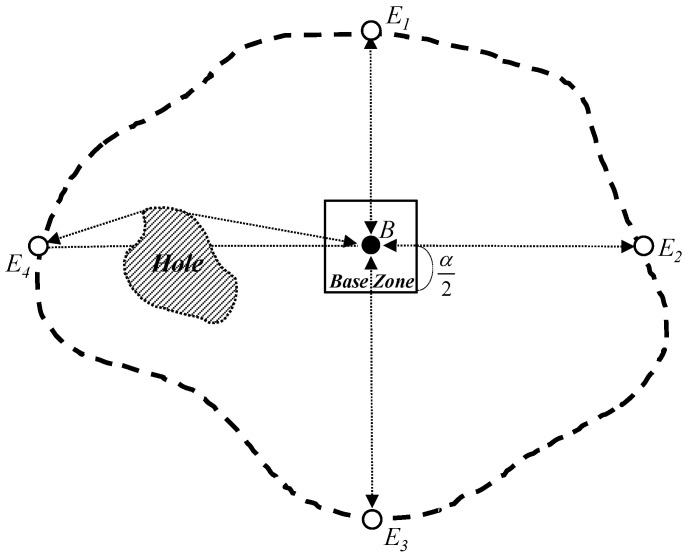
The Base Node B constructs the Base Zone with the length α and then collects the locations of four edge nodes E1, E2, E3, and E4 on horizontal and vertical lines from itself.

**Figure 3 sensors-25-04078-f003:**
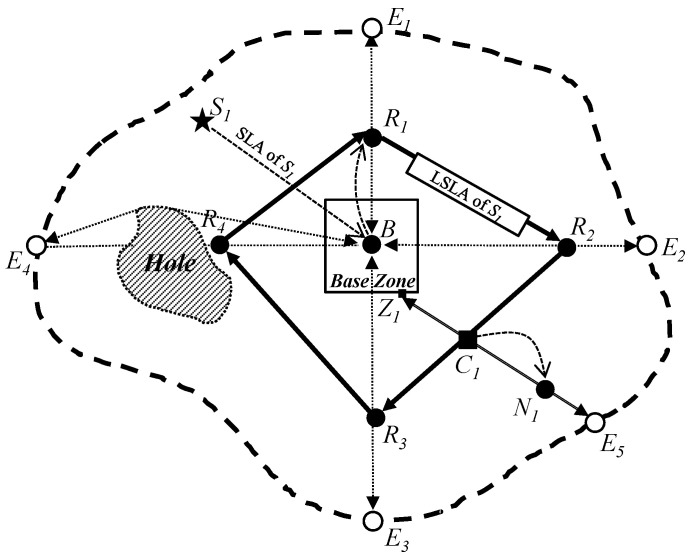
Our energy-balanced quorum-based sink location service scheme: a sink S1 constructs a quadrangle LSLA quorum connected by four random points R1, R2, R3, and R4. Then, a source N1 constructs a line SLQ quorum connected by a random point Z1 on the perimeter of the Base Zone inside the LSLA quorum and a random point E5 on the network perimeter outside the LSLA. Thus, the LSLA and the SLQ has a crossing point C1 between them.

**Figure 4 sensors-25-04078-f004:**
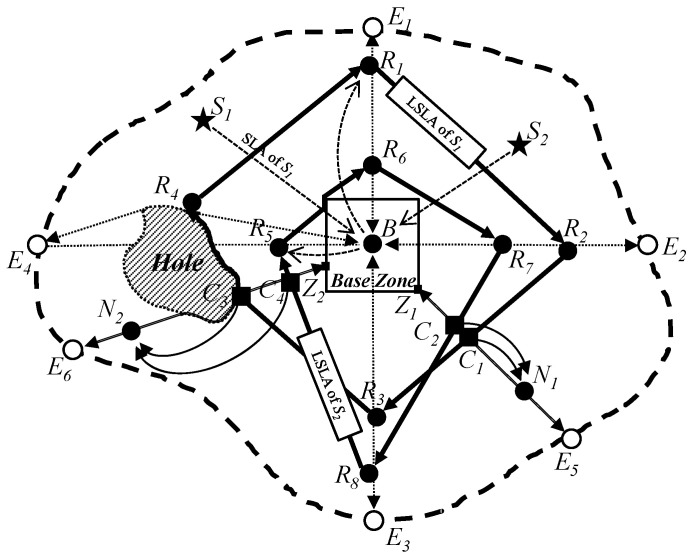
The proposed scheme can handle holes and irregular network boundaries and support multiple sinks and sources.

**Figure 5 sensors-25-04078-f005:**
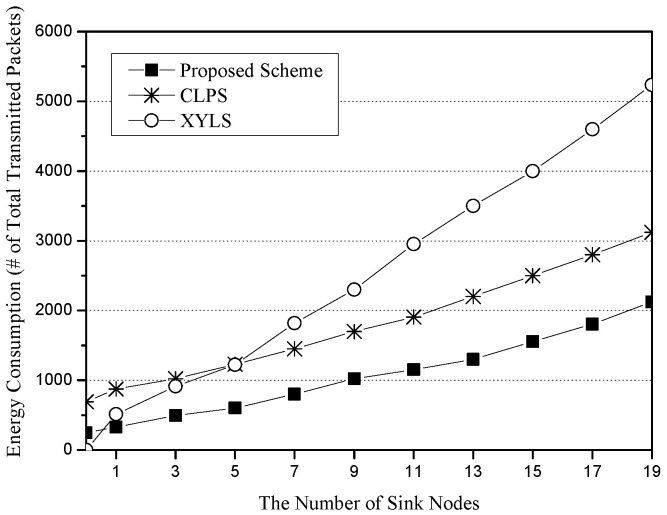
The energy consumption (i.e., the number of total transmitted packets) for the number of sink nodes.

**Figure 6 sensors-25-04078-f006:**
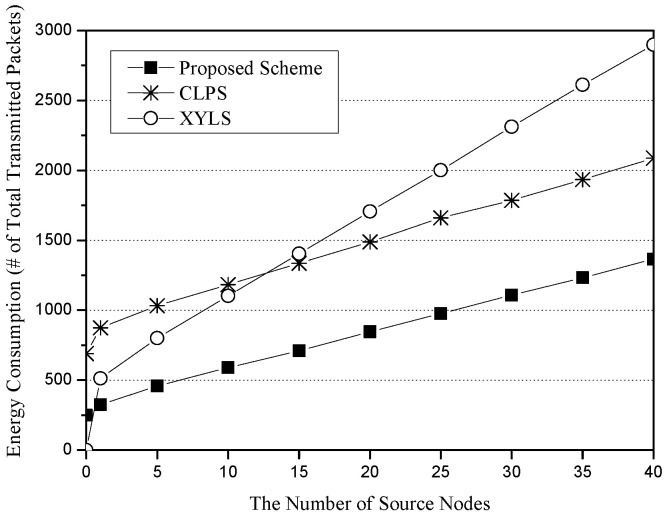
The energy consumption (i.e., the number of total transmission packets) for the number of source nodes.

**Figure 7 sensors-25-04078-f007:**
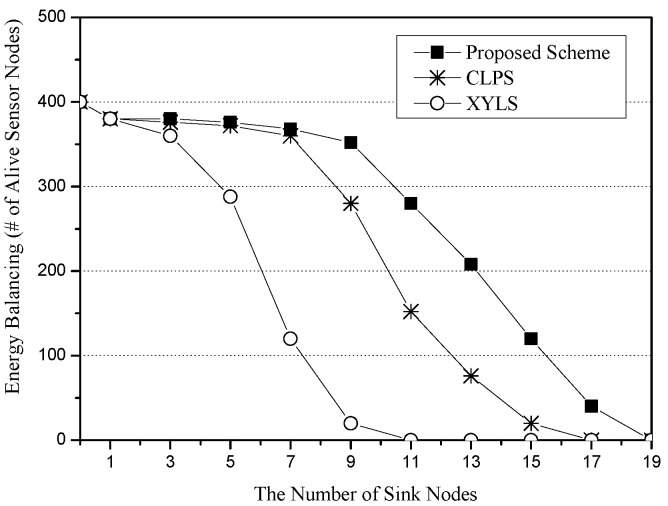
Energy balancing (i.e., the number of alive sensor nodes) for the number of sink nodes.

**Figure 8 sensors-25-04078-f008:**
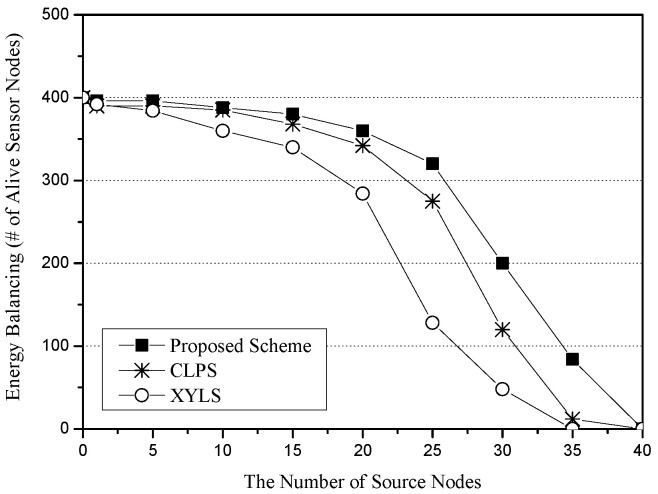
Energy balancing (i.e., the number of alive sensor nodes) for the number of source nodes.

**Figure 9 sensors-25-04078-f009:**
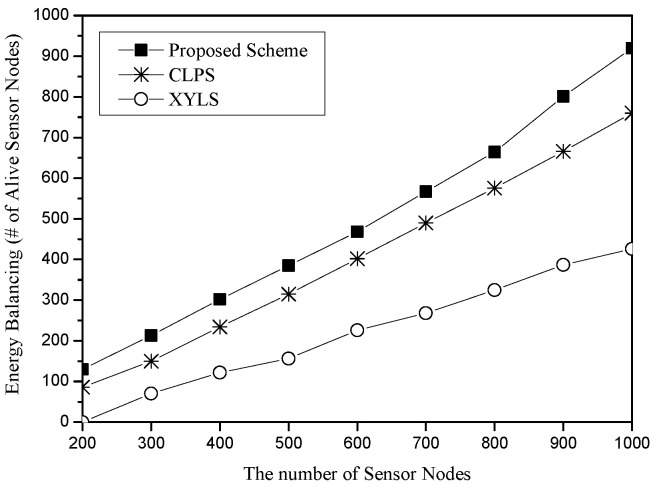
Energy balancing (i.e., the number of alive sensor nodes) for the number of sensor nodes.

**Figure 10 sensors-25-04078-f010:**
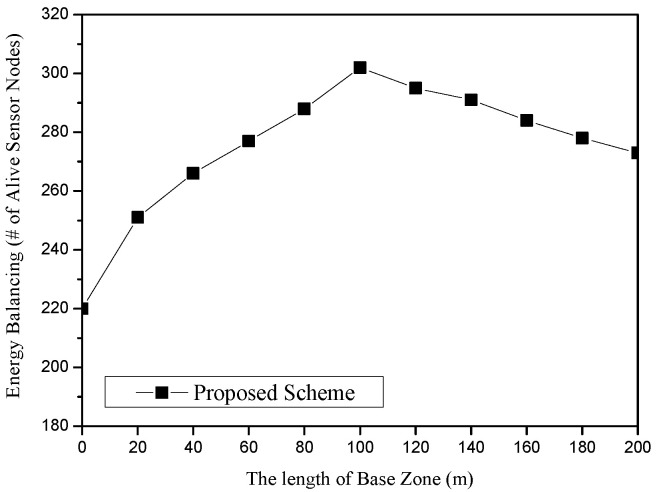
Energy balancing (i.e., the number of alive sensor nodes) for the length of the Base Zone in the proposed scheme.

**Figure 11 sensors-25-04078-f011:**
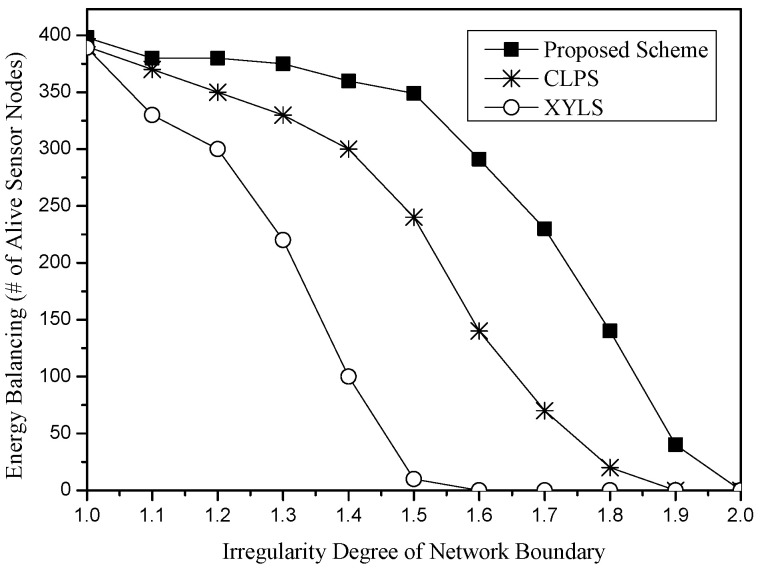
Energy balancing (i.e., the number of alive sensor nodes) for the irregularity degree of the network perimeter.

**Figure 12 sensors-25-04078-f012:**
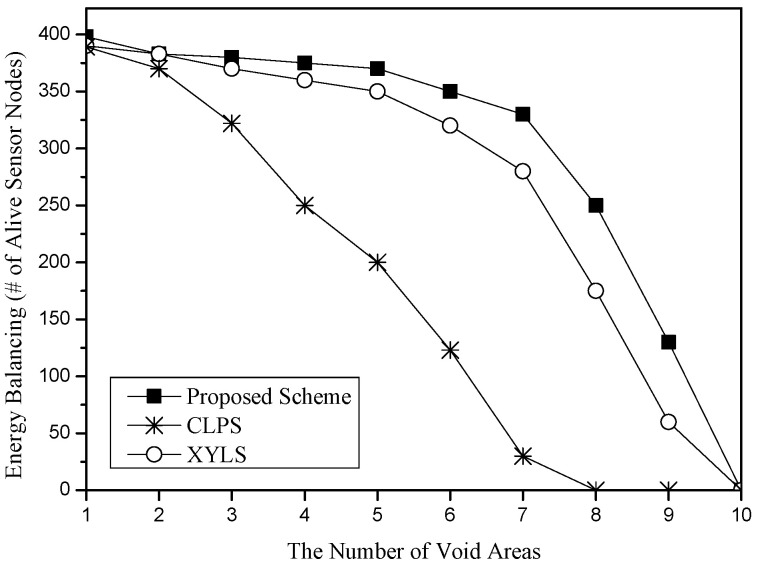
Energy balancing (i.e., the number of alive sensor nodes) for the number of voids.

**Table 1 sensors-25-04078-t001:** Algorithm parameters.

Parameter	Description
*S*	Sink node initiating the sink location announcement (SLA)
BaseNode	The central node used to compute quadrilateral destinations
α	Length parameter defining the size of the Base Zone
packet	Packet containing either LSLA or SLQ information
*X*	Current node processing a packet
nextDest	Next destination node in SLA forwarding
dests	List of destination nodes forming the SLA quadrilateral
rem	Remaining destinations to be processed in SLA
*N*	Sensor node detecting an event and initiating SLQ
interest	Type or content of the event of interest
N_loc	Location of the node that initiates SLQ
S_loc	Location of the sink node initiating SLA
LSLA_info	Stored information of the received LSLA packet
SLQ_info	Stored information of the received SLQ packet
reply	Sink Location Reply (SLR) message sent to the querying node
BaseZonePerimeter	Target location at the perimeter of the Base Zone
NetworkEdge	Target location at the edge of the entire network

**Table 2 sensors-25-04078-t002:** Comparative performance of XYLS, CLPS, and the proposed scheme. All values represent mean ± standard deviation over 100 simulation runs.

Metric	XYLS	CLPS	Proposed Scheme
Energy Consumption (Total # of Transmitted Packets)	17,500 ± 520	13,000 ± 430	9000 ± 350
Energy Balancing (# of Alive Sensor Nodes)	120 ± 15	180 ± 12	230 ± 10

## Data Availability

The original contributions presented in this study are included in the article. Further inquiries can be directed to the corresponding author.

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
