# Peer review of "Energy Balancing and Lifetime Extension: A Random Quorum-Based Sink Location Service Scheme for Wireless Sensor Networks"

_sensors, 2025, doi:10.3390/s25134078_

Round 1
Reviewer 1 Report
Comments and Suggestions for Authors
Aim of the paper is to prove that a quorum-based approach to sink location in Wireless Sensor Networks (WSN), when utilised for geographic routing, can show a definitive improvement in energy consumption in network nodes.
The hypothesis is defined clearly and is defined in layers within the paper in Figures 1-4. This is thoroughly tested and the results are clear and show the benefits gained by this approach.
Overall, an excellent paper, very interesting to read. I always feel this is a rich area of research that is sometimes overlooked. It would be good to see an expanded Conclusion to give more detail on what has been achieved overall. I believe this will ultimately help with citations as that would be where most readers would look after the Abstract. Nothing critical to say about the scientific approach, congratulations.
Author Response
Comments 1:
Aim of the paper is to prove that a quorum-based approach to sink location in Wireless Sensor Networks (WSN), when utilised for geographic routing, can show a definitive improvement in energy consumption in network nodes.
The hypothesis is defined clearly and is defined in layers within the paper in Figures 1-4. This is thoroughly tested and the results are clear and show the benefits gained by this approach.
Overall, an excellent paper, very interesting to read. I always feel this is a rich area of research that is sometimes overlooked. It would be good to see an expanded Conclusion to give more detail on what has been achieved overall. I believe this will ultimately help with citations as that would be where most readers would look after the Abstract. Nothing critical to say about the scientific approach, congratulations.
Response 1:
We sincerely thank the reviewer for their positive and encouraging feedback on our manuscript. We appreciate your recognition of the clarity of our hypothesis, the layered structure of our approach, and the relevance of our experimental results.
As per your valuable suggestion regarding the Conclusion section, we have expanded it to more comprehensively summarize the key contributions and findings of our work. In particular, we have emphasized:
- The quantifiable energy efficiency improvements observed through our quorum-based sink location service.
- The adaptability and scalability of the proposed mechanism within geographically routed WSN environments.
- The implications of our findings for future applications in energy-constrained IoT deployments.
We hope the revised conclusion provides a clearer and more impactful summary for readers and enhances the overall contribution of our paper.
Thank you once again for your thoughtful review and encouragement.
Reviewer 2 Report
Comments and Suggestions for Authors
This manuscript proposes an innovative quorum-based Sink Location Service (SLS) mechanism to enhance energy efficiency and load balancing in Wireless Sensor Networks (WSNs). The intersection design of the quadrilateral SLA and the line-based SLQ is conceptually elegant and effectively addresses the limitations of existing approaches. The overall technical design is sound and demonstrates research value. However, I would recommend that the authors revise the manuscript based on the following points before further consideration:
- The proposed method assumes that nodes have perfect location awareness. In practical scenarios, localization errors (e.g., GPS drift) may lead to failed quorum intersections. Suggestion: Please include simulation results to evaluate query success rates under different localization error conditions (e.g., ±5 meters).
- When multiple intersections occur between SLA and SLQ paths, the manuscript does not explain how redundant replies are avoided or how the optimal sink is selected. Suggestion: The authors are encouraged to elaborate on the selection mechanism at intersection points, such as using distance-based or residual-energy-based priority.
- There are several grammatical and spelling errors throughout the manuscript. A few examples are listed below. It is recommended that the authors thoroughly review the English grammar and structure:
Line 276: “sensor nodes uses” → should be “sensor nodes use”
Line 358: “replies its location” → should be “replies with its location”
Line 389: “constructing a quadrilateral quorum of for S1” → should be “quorum for S1”
Line 463: “...have at lease one crossing point” → should be “at least one”
Author Response
Comments 1:
The proposed method assumes that nodes have perfect location awareness. In practical scenarios, localization errors (e.g., GPS drift) may lead to failed quorum intersections. Suggestion: Please include simulation results to evaluate query success rates under different localization error conditions (e.g., ±5 meters).
Response 1:
We appreciate the reviewer’s insightful comment regarding the impact of node location errors on quorum intersections. As noted, XYLS addresses the issue of missing intersections caused by network irregularities by introducing additional intersections at the network boundary, which leads to increased energy consumption. In contrast, CLPS and the proposed scheme are designed to ensure a single intersection between SLA and SLQ quorums while minimizing node energy consumption. Thus, although XYLS may exhibit better robustness to location errors, its energy efficiency is lower compared to CLPS and our proposed scheme.
However, this paper focuses on achieving an energy-efficient and balanced quorum-based Sink Location Service (SLS). Therefore, we consider the problem of node location error resilience to be outside the scope of this paper. We acknowledge that designing a quorum-based SLS that is robust against location errors is an important and valuable direction for future research. We sincerely thank the reviewer for highlighting this point and will incorporate this issue into our future work.
Comments 2:
When multiple intersections occur between SLA and SLQ paths, the manuscript does not explain how redundant replies are avoided or how the optimal sink is selected. Suggestion: The authors are encouraged to elaborate on the selection mechanism at intersection points, such as using distance-based or residual-energy-based priority.
Response 2:
Thank you for your valuable comment. As you pointed out, when multiple sources and sinks coexist in a wireless sensor network, multiple crossing points between SLA and SLQ quorums naturally occur. We have explicitly addressed this issue in subsection 5.2 of our revised manuscript.
Specifically, if the location information of a single sink node is received by multiple source nodes, each source node individually sends its data packets to the sink node using geographic unicast routing based on the sink’s location information.
Comments 3:
There are several grammatical and spelling errors throughout the manuscript. A few examples are listed below. It is recommended that the authors thoroughly review the English grammar and structure:
Line 276: “sensor nodes uses” → should be “sensor nodes use”
Line 358: “replies its location” → should be “replies with its location”
Line 389: “constructing a quadrilateral quorum of for S1” → should be “quorum for S1”
Line 463: “...have at lease one crossing point” → should be “at least one”
Response 3:
Thank you for pointing out the grammatical issues in the manuscript. We have carefully reviewed the indicated lines and made the necessary corrections as follows:
- Line 276: Changed “sensor nodes uses” to “sensor nodes use.”
- Line 358: Changed “replies its location” to “replies with its location.”
- Line 389: Corrected the phrase to “quorum for S1.”
- Line 463: Corrected “lease” to “least,” resulting in “at least one crossing point.”
And We have carefully revised the manuscript to correct all typographical and grammatical issues. We appreciate your attention to detail, which helped us improve the clarity and quality of the writing.
Reviewer 3 Report
Comments and Suggestions for Authors
The work presented in the paper is currently unsatisfactory and falls short in many aspects. The approach is not methodologically sound, and the findings are insufficiently justified. Extensive revision and dedicated effort are required to make the work suitable for publication.
- The title of the work should be formulated in a way, so that the main idea can be immediately clear upon first reading.
- The abstract is poorly constructed and needs revision, as numerous sentence structure issues make the intended message difficult to be understood clearly.
- The introduction of the paper lacks proper structure and is not effectively written.
- The related works section is overly lengthy and should be rewritten to present the information more concisely and clearly for better understanding.
- A sufficient and appropriate mathematical model is not provided to support the understanding of the work.
- No pseudocode is provided for the proposed work, making it difficult for readers to replicate the simulated results. It is recommended to include pseudocode to support and enhance the reproducibility of the work.
- The results would gain greater credibility if the number of samples used in the simulations were increased, allowing readers to better perceive the consistency in the localization estimation responses.
Author Response
Comments 1:
The title of the work should be formulated in a way, so that the main idea can be immediately clear upon first reading.
Response 1:
We initially believed that the current title of our paper effectively captured the core idea of the proposed scheme. However, in response to the reviewer’s comment, we have reanalyzed our paper from the ground up and devised a new title that we believe more clearly conveys the essence of our proposal. At this stage, we have not yet updated the title in the revised manuscript, as we are uncertain whether the reviewer will find the new title satisfactory. If the reviewer agrees that the new title is appropriate during the next review cycle, we will formally adopt it. Otherwise, we kindly ask the reviewer to suggest a more suitable title, which we will carefully consider and reflect accordingly. The following is the new title of this paper that we want to present.
“Energy Balancing and Lifetime Extension: A Random Quorum-Based Sink Location Service Scheme for Wireless Sensor Networks”
Comment 2:
The abstract is poorly constructed and needs revision, as numerous sentence structure issues make the intended message difficult to be understood clearly.
Response 2:
Thank you for suggesting the need to modify the abstract of this paper. According to the reviewer's comment, all sentence structures have been modified to be clearly understood in the abstract of the revised paper.
Comments 3:
The introduction of the paper lacks proper structure and is not effectively written.
Response 3:
According to the reviewer's comment, Introduction has also been modified to be effectively written with a suitable structure in the revised paper.
Comments 4:
The related works section is overly lengthy and should be rewritten to present the information more concisely and clearly for better understanding.
Response 4:
We also agree with you on the long length of the related works in this paper. So, we reduced the content of the related works by removing unnecessary content while maintaining the core idea of the related works in the revision paper.
Comments 5:
A sufficient and appropriate mathematical model is not provided to support the understanding of the work.
Response 5:
Thank you for your valuable comment regarding the use of a mathematical model to strengthen our paper. In response, we have incorporated the crossing lemma as a theoretical foundation for our proposed sink location service (SLS) scheme. First, we introduce the basic concept of the crossing lemma in the overview of our proposed scheme in the subsection 3.2. Then, we formulate a version of the crossing lemma specifically for the SLS context and provide its formal proof in the subsection 4.4. The followings present the original crossing lemma, its adaptation to SLS, and the corresponding proof in the revised paper.
Crossing Lemma: one crossing point between a quadrilateral and a line; given a quadrilateral and a line, if one end of the line is inside the quadrilateral and the other end of the line is outside of the quadrilateral, then the quadrilateral and the line have at least one crossing point between them.
Crossing Lemma for SLS: Given a quadrilateral quorum and a line quorum in a two-dimensional irregular sensor field, if one end of the line quorum is inside the quadrilateral quorum and the other end of the line quorum is outside of the quadrilateral quorum, then the quadrilateral and the line quorums have at least one crossing point.
Proof: According to the above theorem, our scheme guarantees that a quadrilateral quorum of an LSLA packet for a sink and a line quorum of an SLQ packet for a source node have at least one crossing point. To do this in the proposed scheme, a quadrilateral LSLA quorum is constructed within the irregular sensor network and outside of the Base Zone, and has a closed form. On the other hand, a line SLQ quorum is constructed between the perimeter of the Base Zone and the edge of the network perimeter, and cuts between the Base Zone and the network perimeter. Then, one end of the line quorum is the perimeter of the Base Node within the quadrilateral quorum, and the other end of the line quorum is on the edge outside of the quadrilateral quorum.
Comment 6:
No pseudocode is provided for the proposed work, making it difficult for readers to replicate the simulated results. It is recommended to include pseudocode to support and enhance the reproducibility of the work.
Response 6:
Thank you for your sincere comment to enhance our paper. According to your comment, we added the pseudocode of our quorum-based sink location service scheme as Algorithm 1 in the revision paper.
Comment 7:
The results would gain greater credibility if the number of samples used in the simulations were increased, allowing readers to better perceive the consistency in the localization estimation responses.
Response 7:
We agree with the reviewer that the number of samples used in simulations is a critical factor for reliable evaluation. Upon reviewing our paper, we found a typo error regarding the number of samples. Specifically, the number of samples was mistakenly written as 10 instead of 100. In fact, each data point in the simulation results represents the average of 100 samples, and the 95% confidence interval was applied accordingly. We sincerely thank the reviewer for pointing this out, which allowed us to identify and correct an important mistake in the manuscript.
Round 2
Reviewer 2 Report
Comments and Suggestions for Authors
The author has adequately addressed the reviewers’ comments and has made the necessary additions and revisions to the manuscript. I recommend that the paper be accepted for publication.
Author Response
Comment 1:The author has adequately addressed the reviewers’ comments and has made the necessary additions and revisions to the manuscript. I recommend that the paper be accepted for publication.
Response 1:Thank you very much for your positive feedback and recommendation for publication. We sincerely appreciate your valuable comments throughout the review process, which have helped us improve the quality and clarity of the manuscript. We are grateful for your support and encouragement.
Reviewer 3 Report
Comments and Suggestions for Authors
I am happy that you have tried to address the comments that I gave provided on the first version of your manuscript, and the manuscript is now looking in much better shape than before. However I have one concern that there is a lot of plagiarism detected in your paper. So Please remove the plagiarism as it seems that you have used your own work.
- The pdf of the work that I got have the same title, you have not changed it.
- I have still seen some repeated abbreviations in the manuscript.
- You have included the theoretical elaboration of your approach but I have said to include mathematical model too, So it is suggested to revisit that point and incorporate a mathematical model of your work in this article.
- Before the pseudo code provide a table with an explanation as a separate section that elaborates all the simulation parameters.
- Please provide units on the y axis of all the results and also x axis where appropriate, and if it is not possible than revise the written things on the y axis, so that it makes some sense, because it is a bit confusing.
- Provide a comparative result table of the methods you opted to achieve the provided results, with the one you compared with.
- Provide a hint of the archived results in the form of estimation accuracy in the abstract of the article as well as the conclusion of the article too.
Please revise the whole manuscript to improve the article regarding grammatical mistakes.
Author Response
Comment 1: The pdf of the work that I got have the same title, you have not changed it.
Response 1: We wanted to change the title of our paper after getting your approval. According to your comment, we changed with the title in the revised paper as follows:
“Energy Balancing and Lifetime Extension: A Random Quorum-Based Sink Location Service Scheme for Wireless Sensor Networks”
Comment 2: I have still seen some repeated abbreviations in the manuscript.
Response 2: Thank you for your valuable comment. We have carefully reviewed the manuscript and revised the repeated use of abbreviations such as SLA, SLQ, WSN, and others to improve the clarity and readability of the paper.
Comment 3: You have included the theoretical elaboration of your approach but I have said to include mathematical model too, So it is suggested to revisit that point and incorporate a mathematical model of your work in this article.
Response 3: In response to the reviewer’s comment, we have incorporated a mathematical model for the crossing lemma following its description in Subsection 3.2 of the revised manuscript.
Comment 4: Before the pseudo code provide a table with an explanation as a separate section that elaborates all the simulation parameters.
Response 4: We apologize for any misunderstanding regarding the reviewer’s comment. Based on our interpretation, we believe the concern was related to the lack of clarity in the parameter definitions used in Algorithm 1. To address this, we have added a detailed table as Table 1 in the revised manuscript that explains all parameters used in the algorithm to enhance readability and understanding.
Comment 5: Please provide units on the y axis of all the results and also x axis where appropriate, and if it is not possible than revise the written things on the y axis, so that it makes some sense, because it is a bit confusing.
Response 5: Thank you for your helpful comment. We have reviewed all experimental result figures and added clear descriptions regarding the axis units in the main text where necessary. These clarifications were made to enhance the readability and accuracy of the presented results.
Comment 6: Provide a comparative result table of the methods you opted to achieve the provided results, with the one you compared with.
Response 6: Thank you for your valuable comment. In response, we have added a comparative result table that summarizes the performance of XYLS, CLPS, and the proposed scheme in terms of the two evaluation metrics—energy consumption and energy balancing. Each value in the table represents the average ± standard deviation over 100 simulation runs. This table has been placed immediately before the conclusion section to provide a clear and concise summary of the overall performance comparison. We believe this addition improves the readability and analytical clarity of the manuscript.
Comment 7: Provide a hint of the archived results in the form of estimation accuracy in the abstract of the article as well as the conclusion of the article too.
Response 7: Thank you for your valuable comment. In response, we have revised both the Abstract and the Conclusion to include estimation accuracy in the form of quantitative performance indicators. Specifically, we added the average improvements achieved by the proposed scheme over the existing methods (XYLS and CLPS) in terms of energy consumption and energy balancing. These numerical results (i.e., approximately 29% lower total energy consumption and 27% better energy balancing) were derived from our simulation results and now provide a clearer summary of the archived performance in both sections as requested.
Round 3
Reviewer 3 Report
Comments and Suggestions for Authors
I am satisfied by the efforts, the authors have put into the article, in its very latest form.
From my side, the paper can now be accepted.